# Metamitron Thinning Efficacy of Apple Fruitlets Is Affected by Different Rates, Timings and Weather Factors in New York State

Luis Gonzalez Nieto [1],* , Poliana Francescatto [2], Bruno Carra [3] and Terence Lee Robinson [1]

[1]  Horticulture Section, School of Integrative Plant Science, Cornell University, Geneva, NY 14456, USA; tlr1@cornell.edu
[2]  Valent BioSciences, Libertyville, IL 60048, USA; poliana.francescatto@valentbiosciences.com
[3]  Instituto Nacional de Investigación Agropecuaria (INIA), Sistema Vegetal Intensivo, Estación Experimental INIA Las Brujas, Ruta 48 Km 10, Rincón del Colorado, Canelones 70006, Uruguay; bcarra@inia.org.uy
*  Correspondence: lg579@cornell.edu

**Abstract:** Precision chemical thinning is the most common method of thinning apple fruitlets because it requires little time and is cost-effective. The aims of the current study were I.- to investigate the effect of the application of metamitron at different rates on 'Gala' apples; II.- to determine which fruit diameters were most sensitive to metamitron spray at several rates (between 180 and 500 ppm); and III- to identify the key environmental factors that explain Metamitron efficacy on a year-to-year basis. Eighteen trials were conducted over seven seasons, from 2015 to 2022 in 'Gala' apple orchards in Geneva (New York State). Metamitron was applied at different rates between 180 and 500 ppm, and the timing of the application was between petal fall (4.5 mm) and 18.5 mm fruit size. In each of the studies and years, the effect of meteorological parameters was evaluated. Our results suggest that a linear rate effect was observed in all trials, but that there were differences between the slopes of the regression every year because chemical thinning efficacy was variable year to year. The maximum metamitron efficacy was between 9.5 and 11 mm king fruit diameter; however, metamitron showed thinning efficacy at all phenological stages, from petal fall to 18.5 mm in 'Gala' apples. Our results suggest that the important meteorological factors affecting thinning efficacy were temperature and solar radiation on the day of application and for the next 6 days. The solar radiation after application of metamitron was the most important meteorological factor. Fruit drop caused by metamitron increased with low solar radiation. The minimum and maximum temperatures were also important factors in determining metamitron efficacy. A high minimum temperature (during the night) increased the fruit drop caused by metamitron and the maximum temperature during the day showed a negative correlation with the efficacy of metamitron.

**Keywords:** metamitron; brevis; solar radiation; minimum temperature; maximum temperature; thinning efficacy





## 1. Introduction

Crop load management is the single most important yet difficult management strategy involved in determining the annual profitability of apple orchards [1]. Apple trees generally produce too many flower clusters and fruit for an optimum crop load [2]. Only about 3–10% of the initial fruit population should be carried to harvest to optimize crop value and promote annual bearing. When the fruit set is too high, low-quality fruit is produced and biennial bearing is induced [3]. Chemical thinning is the most common method of crop load management because it is time- and cost-effective. However, chemical thinning efficacy is variable year to year because it is affected by weather conditions and cultivar [2,4–8]. Errors in chemical thinning applications can result in over-thinning, with a low crop load, excessively large fruit size and reduced crop value, or in under-thinning, resulting in small fruit size and lower total crop value.

Metamitron is a photosynthetic inhibitor [9,10] that belongs to the triazinone family of herbicides. It has a different mode of action to other traditional chemical thinning products. Metamitron disrupts the photosynthetic apparatus after application and acts by blocking electron transfer between primary and secondary quinones of PSII [11]. This inhibition reduces carbohydrate production by the tree, producing a carbohydrate deficit in the tree and resulting in carbohydrates being sent to shoots rather than to fruit [12]. The carbohydrates that are sent to fruit are directed to the dominant king fruit, while the smaller fruitlets stop growing and abscise [12].

The moment of application [13] and the rate are key factors in the use of plant growth regulators. Metamitron can be applied to young fruitlets, with king fruit diameters ranging between 6 and 20 mm [13–18]. Optimum application rates may vary from 165 to 330 mg $L^{-1}$, depending on the variety, as leaf sensitivity to the chemical differs according to the cultivar [2,7].

The abscission of fruitlets induced by thinning agents is a complex interaction between environmental conditions, cultivar, fruit size and tree vigor [19]. Temperature, humidity and solar radiation levels are also important factors that need to be taken into account. Byers [6] reported that low-light conditions and periods of high night-time temperatures favor the abscission of fruitlets. Such conditions stimulate a carbohydrate deficit in the fruit. The tree favors the carbohydrate demand, and hence the growth, of shoots over that of fruit, resulting in fruitlet sensitivity to chemical thinning [20,21].

Thus, the maximum efficacy of a chemical thinner depends on the diameter of the developing fruit, application rate, cultivar, and climatology [6]. The aims of the current study were I.- to investigate the effect of application of metamitron at different rates on 'Gala' apples; II.- to determine which fruit diameters are most sensitive to metamitron applications with low, medium and high spray rates (between 180 and 500 ppm); and III.- to identify the key environmental factors that explain metamitron efficacy on a year-to-year basis.

## 2. Materials and Methods

### 2.1. Plant Material and Experimental Design

Eighteen trials were conducted over eight seasons, from 2015 to 2022, in apple orchards (*Malus* × *domestica* Borkh.) at Cornell University's Agritech Campus in Geneva, New York State. Mature (between 10 and 20 years) and uniform 'Gala' apple trees grafted on M.9 rootstock were selected in terms of bloom density and growth. The training system was tall spindle trees being planted at high density (between 1400 trees $ha^{-1}$ and 2240 trees $ha^{-1}$). All orchards were managed according to the standards normally used in commercial apple orchards in the region. The trees were irrigated and fertilized using a drip irrigation system.

All trials were arranged in a randomized block design with five replicates of three uniform trees per experimental unit. The central tree of each elemental plot was used for the trial assessments.

### 2.2. Study Rate, Timings and Weather Conditions

The trials tested the efficacy of the chemical thinner metamitron. Metamitron was sprayed at different rates between 180 and 500 ppm, and an untreated control was included in each study. Water volume rates were calculated according to the tree row volume methodology [22]. The timing of application was between petal fall (4.5 mm) and 18.5 mm and was determined by measuring king fruit diameter. All treatments were sprayed at 1000 L $ha^{-1}$ of water volume. Table 1 shows the year of each field trial, timings and spray rates.

**Table 1.** Year, timing (mm) and rate (ppm) of application in each experiment.

| Year | Timing (mm) | Rate (ppm) | | | | |
|---|---|---|---|---|---|---|
| | | 0 | 180–200 | 280–300 | 350–375 | 400–500 |
| 2015 | Petal Fall (4.5 mm) | X | X | X | | X |
| | 10 mm | X | X | X | | X |
| 2016 | Petal Fall (4.5 mm) | X | X | | X | X |
| | 15 mm | X | X | | X | X |
| 2017 | Petal Fall (4.5 mm) | X | X | | X | X |
| | 9 mm | X | X | | X | |
| | 13.5 mm | X | X | | X | X |
| 2018 | Petal Fall (4.5 mm) | X | X | | X | X |
| | 9 mm | X | | X | X | X |
| | 10 mm | X | X | | X | X |
| | 15 mm | X | X | X | X | X |
| 2019 | 12 mm | X | X | X | | X |
| 2021 | 18 mm | X | | X | | X |
| 2022 | 11 mm (1) | X | X | X | X | |
| | 11 mm (2) | X | | X | X | |
| | 14 mm | X | | X | X | |
| | 18.5 mm | X | X | X | X | |

### 2.3. Weather Conditions

Meteorological data were collected from a weather station 0.5 km from the trial site and stored on the NEWA website (NEWA, 2018). The meteorological parameters that we evaluated are shown in Table 2. The weather data showed gaps in the data, with data interpolated specifically for temperatures because the sensors did not work on some days between 2015 and 2018. When the data showed gaps, we did not use the interpolated data for the analysis. Table 2 also shows the period used to calculate the average value.

**Table 2.** Weather data evaluated and period of day used to calculate the average value.

| Weather Data Evaluated | Period |
|---|---|
| Solar radiation (MJ/m$^2$) | 08:00 to 20:00 |
| Daytime temperature (°C) | 08:00 to 20:00 |
| Maximum temperature (°C) | 00:00–24:00 |
| 24 h temperature (°C) | 00:00–24:00 |
| Minimum temperature (°C) | 00:00–24:00 |
| Night-time temperature (°C) | 21:00 h up to 7:00 of the next day |

For each trial, we evaluated the effect of each weather variable over the period from −7 days prior to the treatment date up to +7 days after the treatment date. This resulted in an evaluation of the effect of weather factors spanning a period of 16 days around the spray date of treatments. We also evaluated the effect of averages for each variable using a series of moving averages with windows of different widths (between 2 and 16 days), both before and after metamitron sprays. Figure 1 shows an example of a 3-day window for each weather variable.

### 2.4. Thinning Efficacy

The thinning efficacy of each metamitron treatment was defined as the number of fruits at harvest by each application treatment related to the number of fruits at harvest in the untreated control, expressed as a percentage.

$$Fruit\ drop\ (\%) = 100 - \left( \frac{Fruits\ per\ tree\ in\ the\ treatment\ with\ Metamitron}{Fruits\ per\ tree\ in\ Untreated\ Control} \times 100 \right)$$

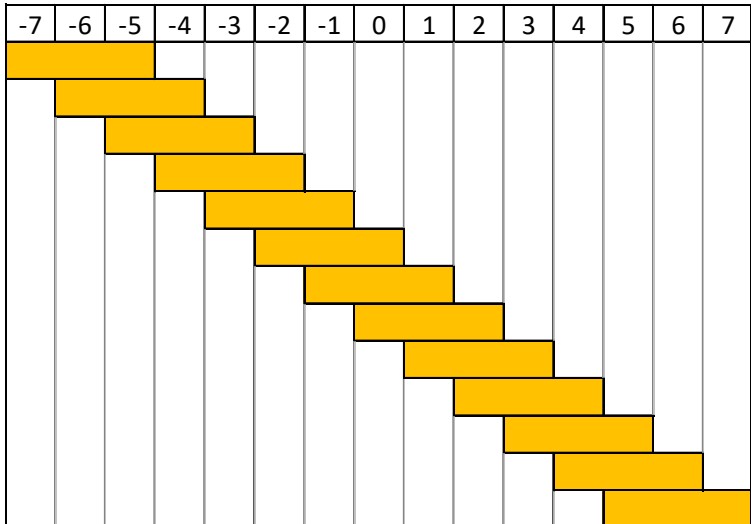

**Figure 1.** Example of a 3-day window of average weather variables.

### 2.5. Statistical Analysis

The effect of chemical rate was analyzed for each experiment using linear regression between the chemical rate and fruit drop (%), averaged over all repetitions in each experiment. A quadratic regression analysis was used to determine the effect of spray timing (fruit development stage) on fruit drop (%).

Multivariate correlation analysis was used to determine the weather window which had the greatest effect on thinning efficacy for each weather variable. The effect of each weather variable during the best weather window (highest r value) was analyzed using linear regression analysis. This analysis was only performed on the weather parameters that showed a highly significant effect on thinning efficacy. These were solar radiation, maximum temperature, and minimum temperature. Linear regression was performed between fruit drop (%) and solar radiation and maximum temperatures. Quadratic regression was performed between fruit drop (%) and minimum temperatures.

## 3. Results

### 3.1. Effect of Rate of Metamitron

All metamitron rates, timings and years showed higher fruitlet drop (%) in comparison with the untreated control. There was a linear rate effect in all trials, with an increase in the rate of metamitron resulting in an increase in fruit drop (%) and the thinning efficacy of metamitron (Figure 2). However, the efficacy of metamitron was different every year. There were differences in the slopes of the regression between experiments and years. The highest slopes were from metamitron sprays at 9 mm fruit size in 2017 and at 13.5 mm fruit size in 2017. The lowest slopes were from sprays at 9 mm fruit size in 2016 and at 18 mm fruit size in 2021 (Figure 2).

### 3.2. Effect of Timing of Application

There was a significant correlation between the timing of metamitron spray and fruit drop (%) for each rate range, except for in the lower range (180–200 ppm) (Figure 3). When the rate was low, there was a higher variability in the efficacy of metamitron. The maximum efficacy was between 9.5 and 11 mm in each of the rate ranges evaluated (Figure 3). However, metamitron showed efficacy at all timings between petal fall and 18.5 mm. When fruit size was greater than 18 mm, the efficacy of metamitron was lower (Figure 3).

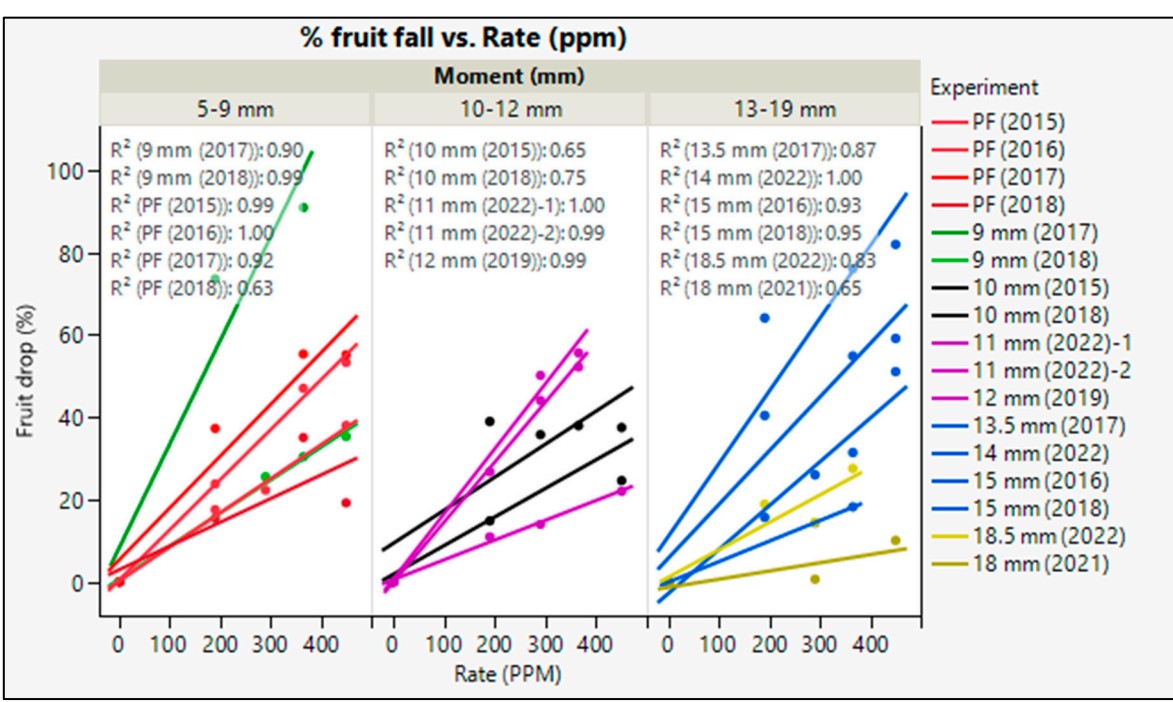

**Figure 2.** Relationships between rate of metamitron (ppm) and fruit drop (%) between 2015 and 2022 in 'Gala' trees in Geneva NY, USA. Each dot represents the average of 5 repetitions.

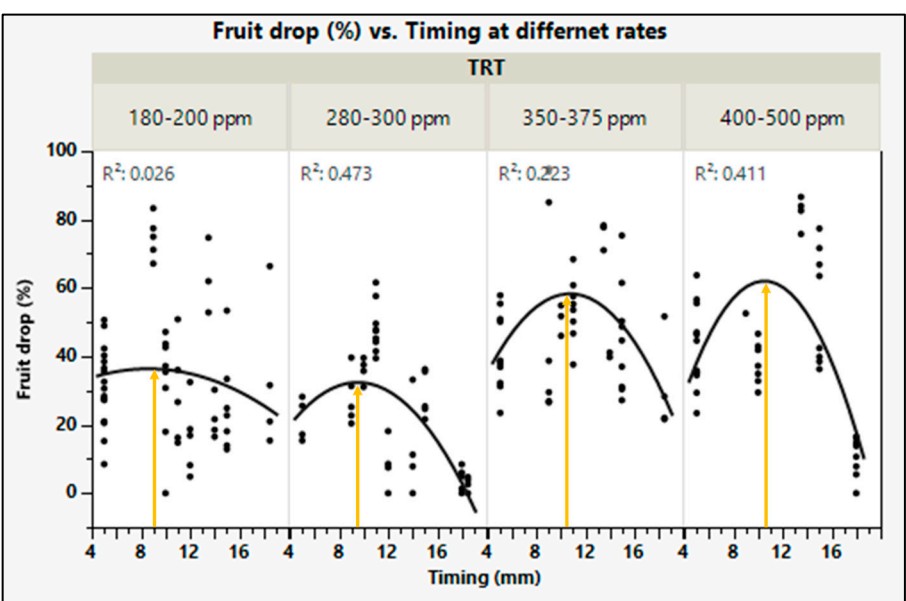

**Figure 3.** Relationships between the timing of metamitron spray (fruit diameter—mm) and fruit drop (%) between 2015 and 2022 in 'Gala' trees in Geneva NY, USA. Timing was determined using king fruit diameter. Each dot represents one tree, and the yellow arrow indicates the maximum efficient point of each rate range.

### 3.3. Effect of Weather Conditions

The multivariate analysis showed significant correlations between solar radiation, maximum and minimum temperatures, and metamitron efficacy. All other weather variables in all of the time windows we evaluated showed r values lower than 0.25 (data not presented). The best correlation between any weather variable and metamitron efficacy was with solar radiation in the window from 1–6 days after the application of metamitron spray. The minimum and maximum temperatures showed similar values of correlation with

metamitron efficacy, but lower correlation values than with solar radiation. The analysis of minimum temperatures showed two important windows (3 days, beginning 1 day before spray through 1 day after the spray; and 4 days, beginning 1 day before the spray and ending 2 days after). The window with the greatest effect of maximum temperatures was 10 days, beginning 2 days before spray through 7 days after.

### 3.3.1. Solar Radiation

All rates of metamitron spray showed a significant correlation between solar radiation in the 6 days after spray application and fruit drop (%) (Figure 4). The R2 values of the correlation for all rates of spray were between 0.8 and 0.44. The higher value of R2 was for lower rates of metamitron (Figure 4). Within each rate range, low solar radiation increased the efficacy of metamitron. That is, cloudy days after the application increased the fruit drop caused by metamitron. At the same value of solar radiation, when the rate of metamitron increased fruit drop and thinning efficacy was high (Figure 4).

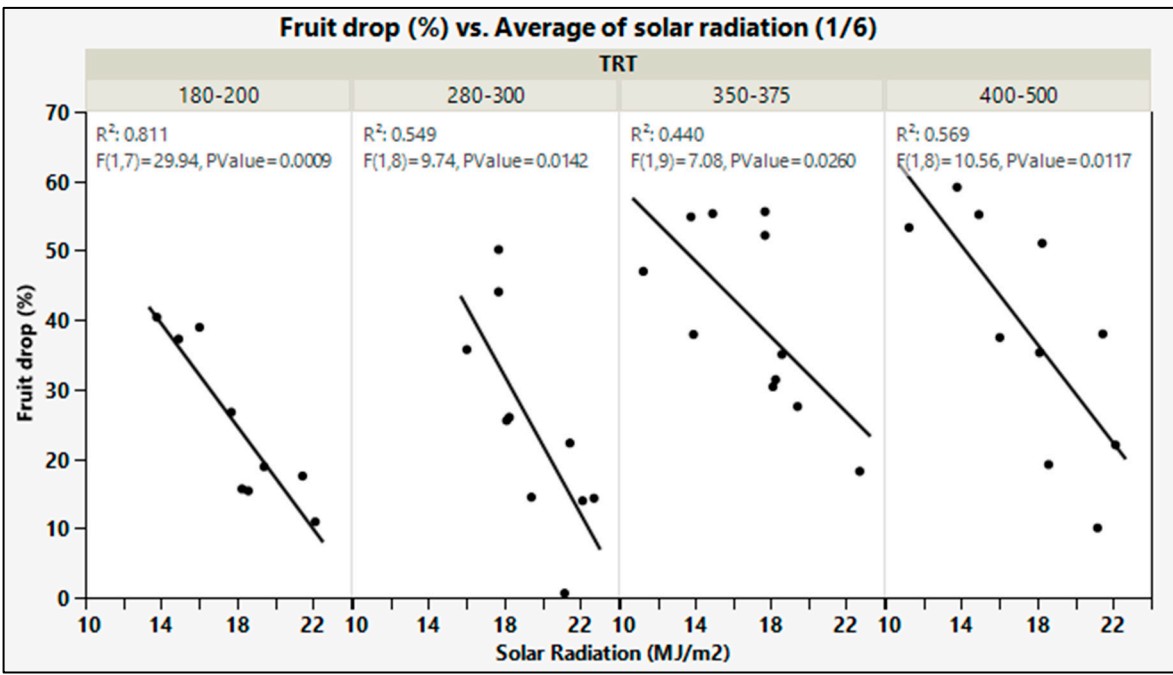

**Figure 4.** Relationships between solar radiation (MJ m$^{-2}$ day$^{-1}$) in the 6 days after application and fruit drop (%) between 2015 and 2022 at different spray rates with 'Gala' trees in Geneva NY, USA. Each dot represents the average of 5 repetitions.

Figure 5 shows that, when solar radiation was lower than 16 MJ m$^{-2}$ day$^{-1}$ in the 6 days after spray application, the efficacy of metamitron was high. When the average of solar radiation was lower than this value, fruit drop caused by metamitron was between 35% and 60%. However, when the solar radiation was between 17 and 18 MJ m$^{-2}$ day$^{-1}$, the efficacy was medium and more variable (between 10% and 55%) (Figure 5). When solar radiation was greater than 18 MJ m$^{-2}$ day$^{-1}$, the efficacy was lower (between 0% and 20%) (Figure 5).

### 3.3.2. Minimum Temperatures

There was a positive and significant correlation between the minimum temperatures over the 3- or 4-day window just before and after spray application, and fruit drop (%) (Figure 6). The values of R2 in all rate ranges of spray were between 0.2 and 0.93. In all rates evaluated, a high minimum (night) temperature increased the efficacy of metamitron. That is, a high temperature during the night increased fruit drop caused by metamitron (Figure 6). There was a rate effect with an increased rate of metamitron at any given

minimum (night) temperature whereby fruit drop increased and thinning efficacy was higher (Figure 6).

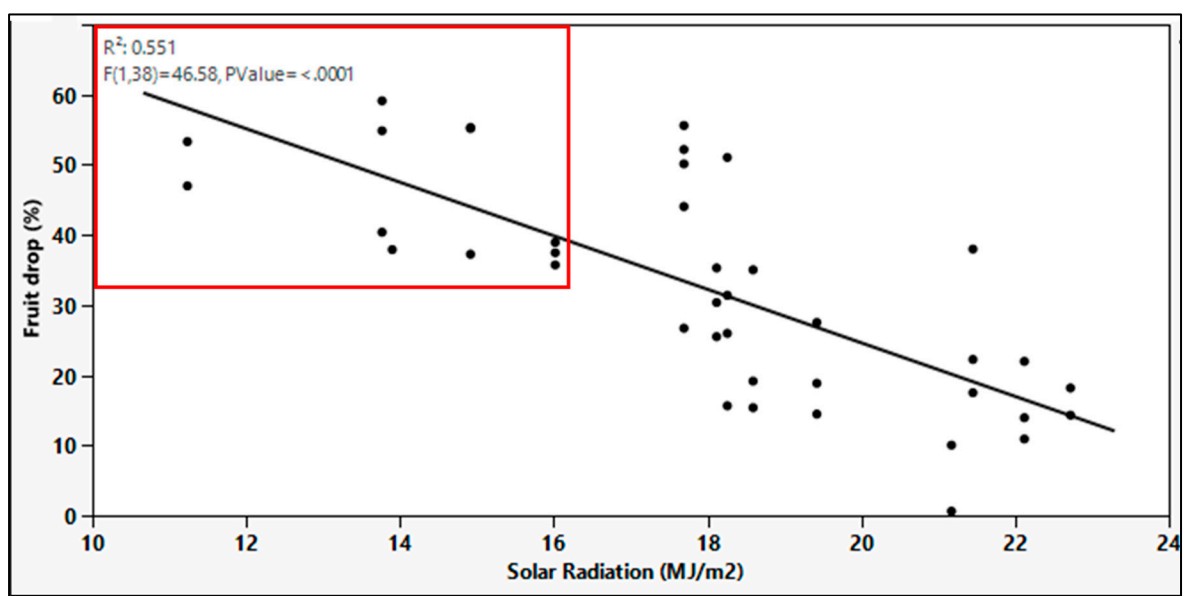

**Figure 5.** Relationships between solar radiation (MJ m$^{-2}$ day$^{-1}$) in the 6 days after application and fruit drop (%) (between 2015 and 2022), with all rates in the same regression with 'Gala' trees in Geneva NY, USA. Each dot represents the average of 5 repetitions. Red square indicates the solar radiation range with higher efficacy of metamitron.

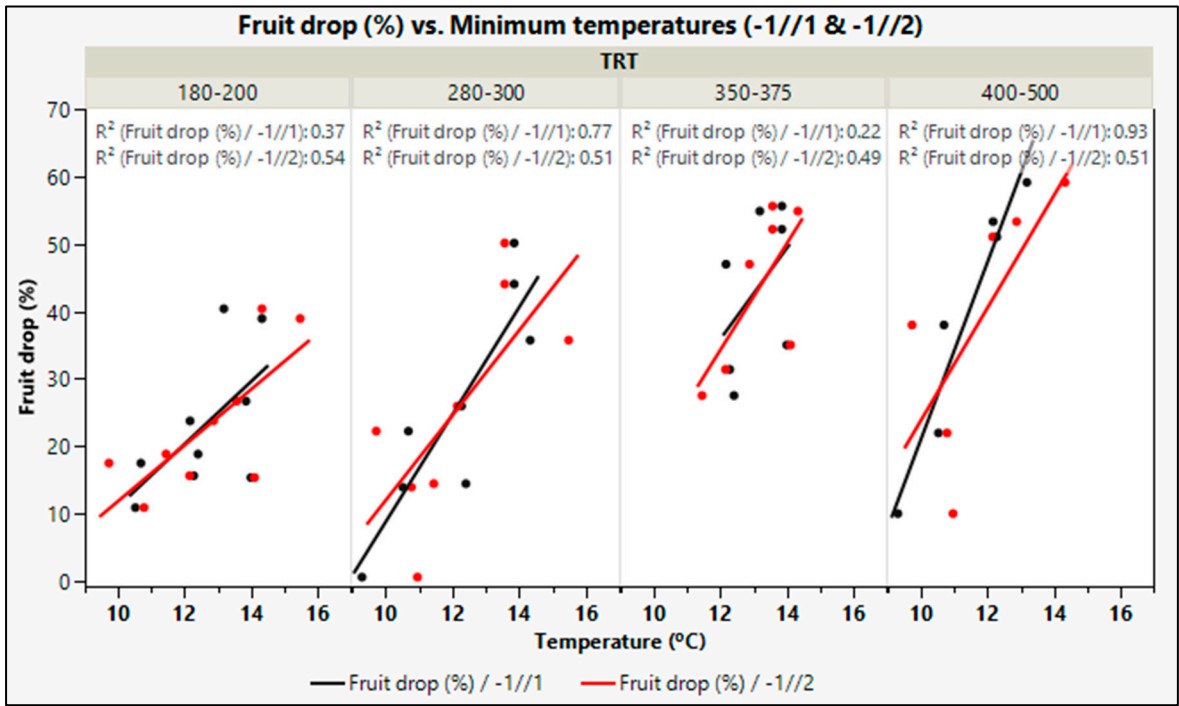

**Figure 6.** Relationships between minimum temperature (°C) over the 3 or 4 days before and after application and fruit drop (%) (between 2015 and 2022) at different rates of spray in 'Gala' trees in Geneva NY, USA. The black line is the regression for the 3-day period, from 1 day before treatment through 1 day after treatment. The red line is the regression for the 4-day period, from 1 day before treatment through 2 days after treatment. Each dot represents the average of 5 repetitions.

Figure 7 shows that the efficacy of metamitron was between 0 and 25% fruit drop when the average minimum temperature over the 3- or 4-day period before and after application was lower than 12 °C. However, when the average minimum temperature was higher than 12 °C, the efficacy was higher than 25% in practically all trials (red box) (Figure 7). When the average minimum temperature was 14.3 °C or higher, the efficiency of metamitron was always between 35 and 60% fruit dop (green box) (Figure 7).

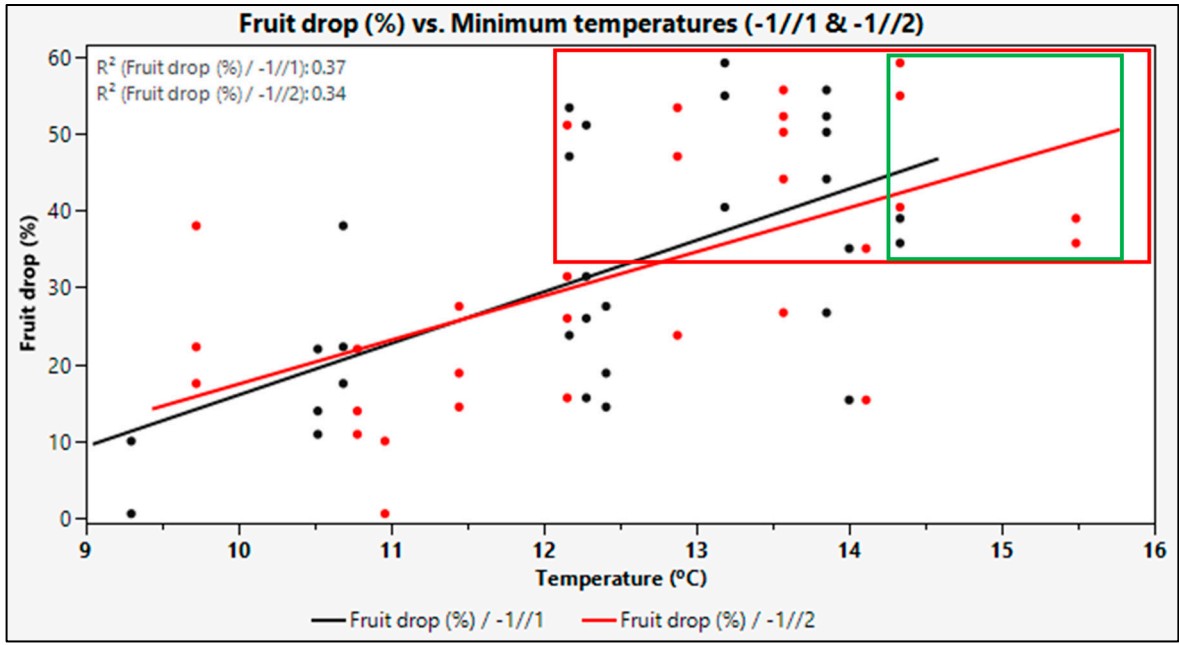

**Figure 7.** Relationships between minimum (night) temperature (°C) over the 3 or 4 days before and after application and fruit drop (%) between 2015 and 2022, with all rates in the same regression with 'Gala' trees in Geneva NY, USA. The black line is the regression for the 3-day period, from 1 day before treatment through 1 day after treatment. The red line is the regression for the 4-day period, from 1 day before treatment through 2 days after treatment. Each dot represents the average of 5 repetitions. The red square indicates the night temperature range with higher efficacy of metamitron. The green square represents the temperature range that always gave high efficiency with metamitron.

3.3.3. Maximum Temperatures (24 h)

There was a negative correlation between maximum temperature over the 10-day period, beginning 2 days before spray through 6 days after, and metamitron thinning efficacy (Figure 8). There was a significant correlation at all rate ranges, except for the range between 280 and 300 mg L$^{-1}$. In this rate range, the correlation was not significant because our results did not show much variability in maximum temperature (between 22 and 25 °C) for the treatments receiving this rate range (Figure 8). Overall, when the average maximum temperature was higher, the efficacy of metamitron was lower.

When the data from all our trials were analyzed together, the efficacy of metamitron was higher when the average maximum temperature over the 10-day period before and after spray application was around 19–20.5 °C (red box) (Figure 9). Within this range of temperatures, the efficacy of metamitron was practically always higher than 55%. When the average maximum temperature was higher than 21 °C, the efficacy was between 0 and 55% (Figure 9). When the average maximum temperature was between 22 °C and 27 °C, the efficacy was lower than 40% (Figure 9).

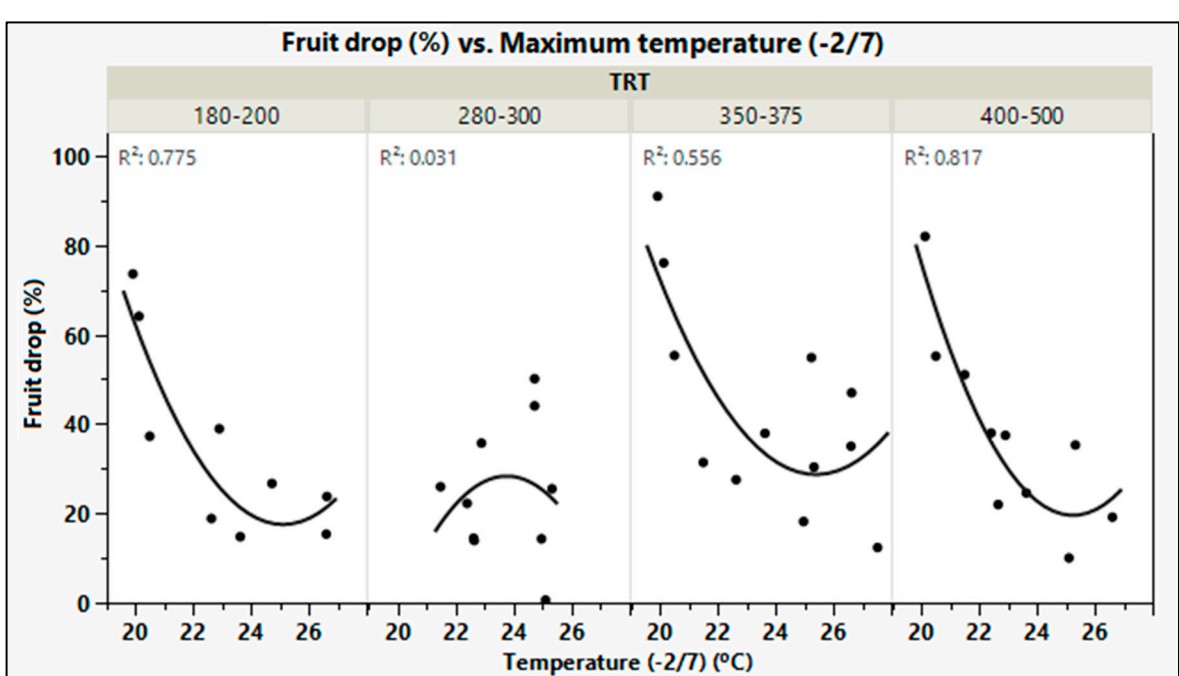

**Figure 8.** Relationships between average daily maximum temperature (°C) over the 10-day period before and after application of metamitron and fruit drop (%) between 2015 and 2022 at different rate ranges of metamitron with 'Gala' apple trees in Geneva NY, USA. Each dot represents the average of 5 repetitions.

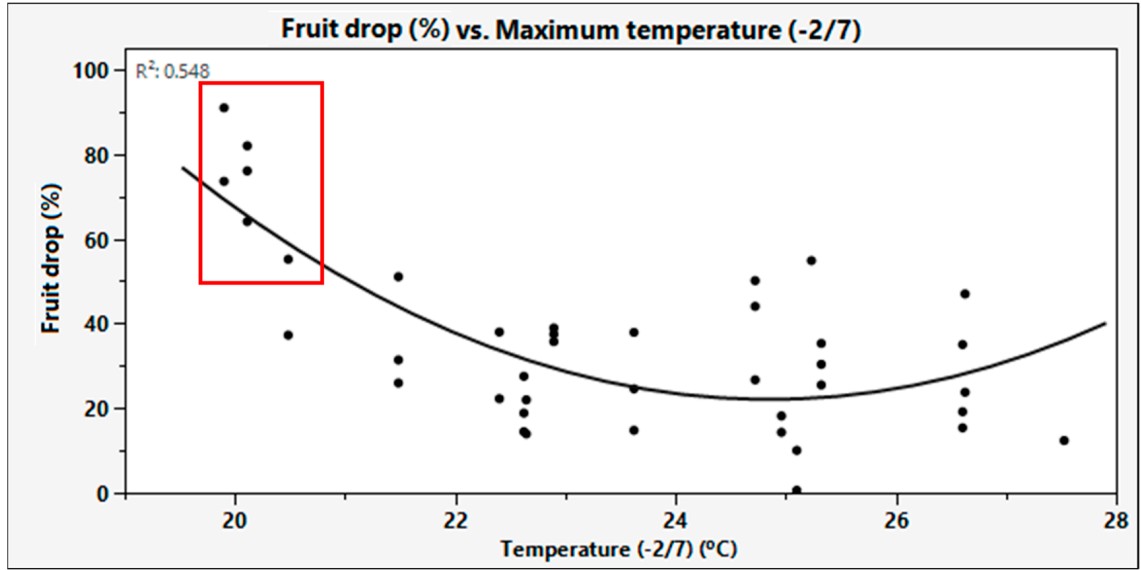

**Figure 9.** The relationship between average maximum temperature (°C) over the 10-day period before and after application of metamitron and fruit drop (%) between 2015 and 2022, with all rates in the same regression with 'Gala' apple trees in Geneva NY, USA. Each dot represents the average of 5 repetitions. The red square indicates the daytime maximum temperature range with higher efficacy of metamitron.

## 4. Discussion

Apple trees generally produce too many flower clusters and too much fruit for an optimum crop load [2]. When the fruit set is too high, low-quality fruit is produced and biennial bearing is induced [3]. Chemical thinning is the most common method of thinning because it is time- and cost-effective [23]. The maximum effectiveness for traditional chemical thinning agents is related to the diameter of the developing fruit, application

rate, cultivar, and weather conditions [6]. Our results with metamitron indicate that its effectiveness as a thinning agent is similarly related to the diameter of developing fruitlets, application rate, and weather conditions, as shown by Byers, et al. [24]. Other studies have previously shown that the thinning efficacy of metamitron depends on the application rate [3,8,13,18,25–27]. Although we observed a linear rate effect of metamitron in all our trials, there were large differences between the slopes of the regression between the trials. This indicates that the chemical thinning efficacy is variable year to year, likely because it is affected by weather conditions [4–6,28].

The phenological stage of fruit development, when chemical thinners are applied, is a key factor for the use of plant growth regulators [13]. In this study, the maximum metamitron efficacy was between 9.5 and 11 mm in terms of king fruit diameter, which varied depending on whether the rate of chemical was high or low. However, this result differs from that of Brunner [25], who obtained maximum efficacy at 12 mm; Reginato et al. [29], who obtained maximum efficacy at 16 mm; and Gonzalez et al. [17], who reported maximum efficacy at 11.5–14 mm. Many authors have reported metamitron efficacy results at different fruit stages between petal fall and 20 mm [13–18,25–27,30,31]. All of these reports, together with results from the present study, show that metamitron thinning efficacy varies significantly with phenological stage from petal fall to 18.5 mm in 'Gala' apples.

The degree of fruit abscission induced by metamitron is highly dependent on environmental factors [4,10,13]. The conditions that lead to carbohydrate deficits in the tree are also associated with heavy drop and easier thinning. These are cloudy conditions, involving a heavy initial set on many weak spurs, stressed trees, photosynthesis inhibitors, natural or imposed low-light periods and high night-time temperatures [21]. The results of the present study agree with this hypothesis, since our results suggest that temperature and solar radiation immediately before and after application were important factors affecting metamitron thinning efficacy.

Our results suggest that solar radiation after the application was the most important weather factor influencing thinning efficacy. This is in line with other reports, which have shown solar radiation to be an important weather factor in explaining chemical thinning efficacy [28,32–36]. In New York State, it is common to have cloudy days during the spring period. These cloudy days cause shortages of carbohydrates required to support fruit growth [37]. Our results suggest that cloudy days (total daily solar radiation less than 16 MJ m$^{-2}$ day$^{-1}$) after metamitron application increased fruit drop. This is in line with the results obtained by Clever [38] in a study with metamitron sprays in apples that used 16 MJ m$^{-2}$ day$^{-1}$ as the critical threshold. This result leads to the hypothesis that metamitron's thinning effect is due to its inhibition of photosynthesis; plus, low solar radiation in the period after application creates a negative carbohydrate balance that causes weaker fruitlets to cease growing and begin abscising.

Temperature plays an important role in apple thinning efficacy with various chemical products [21,39–43]. Previous results concur with the observations of this study, which indicate that minimum and maximum temperatures are important factors in determining metamitron efficacy. Robinson et al. [44] and Jing et al. [45] showed evidence that periods of high night-time temperature stimulate high carbohydrate consumption by enhanced dark respiration and consequently can create a carbohydrate deficit in the plant. These results concur with the observations of this study, in which the high minimum temperature (during the night) increases fruit drop caused by metamitron. Our results suggest three levels of temperature effects: a minimum temperature lower than 12 °C had low thinning efficacy, a minimum temperature between 12 °C and 14 °C had medium thinning efficacy, and a minimum temperature higher than 14 °C had high thinning efficacy. Other authors have reported that the minimum temperature affects metamitron efficacy. Radivojevic et al. [46] showed the positive thinning efficacy of a single application at 6–15 mm of metamitron, with minimum temperatures above of 10 °C. Gonzalez et al. [2] concluded that, when the average night temperature was higher than 14 °C for 4 days after a single application between 7.5 and 13.5 mm, the efficacy of metamitron was high. Clever [38]

estimated that thinning efficacy was higher when the night-time temperature was above 10 °C. However, their results were collected over a different period of days to that of this study (1 day before the spray and 1 or 2 days after the spray).

In contrast to the effect of minimum temperature, the effect of maximum temperature was negatively correlated with the efficacy of metamitron. We explain this result by hypothesizing that, when the maximum temperature was higher, total tree photosynthesis increased and the accumulation of carbohydrates was high and the efficacy of metamitron was lower. Our results suggest that, when the average maximum temperature was below 20.5 °C, the increase in the efficacy of metamitron treatment was due to lower carbohydrate accumulation in the plant because of lower total photosynthesis. Previous reports estimate that the optimum temperature for photosynthesis is around 25 °C in normal conditions [47]. Our results suggest that the accumulation of carbohydrates decreased with lower temperature and, thus, the thinning efficacy of metamitron was greater at lower average daytime temperatures. Moreover, the periods with low daily daytime temperatures were usually on cloudy days with low solar radiation. All these factors increased the efficacy of metamitron.

Overall, our results suggest that the rate of application, timing and the weather conditions were all important in determining the efficacy of metamitron as a chemical thinner. However, it is necessary to better understand the inhibition of the photosynthesis caused by metamitron and the interaction of the inhibition with weather conditions. A better understanding of the physiological effects of metamitron during the thinning period under different weather conditions and the interaction between temperature and photosynthesis and respiration in the presence of metamitron sprays will help to better predict when metamitron will be effective.

## 5. Conclusions

All metamitron rates and spray timings during the evaluated years induced fruit abscission in 'Gala' apple trees. The thinning efficacy of metamitron is rate-dependent, with a linear rate effect observed in all of our trials; however, we found differences between the slopes of the relationship between metamitron rate and thinning efficacy from year to year. Although metamitron showed thinning efficacy at the phenological stages from petal fall to 18.5 mm, the maximum metamitron efficacy was between 9.5 and 11 mm in terms of king fruit diameter, whether the rate of metamitron was low, medium or high.

Our results suggest that the important weather factors affecting metamitron thinning efficacy were temperature and solar radiation. The solar radiation 6 days after the application was the most important weather factor in our study. The fruit drop caused by metamitron increased when daily solar radiation was less than 16 MJ m$^{-2}$ day$^{-1}$. Temperature (minimum and maximum temperature) was also an important factor in determining metamitron efficacy, but had less importance than solar radiation. High minimum temperature (during the night) increased fruit drop caused by metamitron. Our results suggest three levels of efficacy: a minimum temperature lower than 12 °C indicated low efficacy, a minimum temperature between 12 °C and 14 °C indicated medium efficacy, and a minimum temperature higher than 14 °C indicated high efficacy. The maximum daily temperature had an opposite effect, with a negative correlation with the thinning efficacy of metamitron. That is, maximum temperatures below 20.5 °C increased the thinning efficacy of metamitron.

**Author Contributions:** Conceptualization, L.G.N. and T.L.R.; methodology, L.G.N., T.L.R., P.F. and B.C.; software, L.G.N.; validation, L.G.N., T.L.R. and B.C.; formal analysis, L.G.N.; investigation, L.G.N., T.L.R., P.F. and B.C.; resources, L.G.N., T.L.R., P.F. and B.C.; data curation, L.G.N.; writing—original draft preparation, L.G.N.; writing—review and editing, L.G.N., T.L.R., P.F. and B.C.; visualization, L.G.N., T.L.R., P.F. and B.C.; supervision, T.L.R.; project administration T.L.R.; funding acquisition, T.L.R. All authors have read and agreed to the published version of the manuscript.

**Funding:** These studies were supported by grants from ADAMA-USA.

**Data Availability Statement:** Not applicable.

**Acknowledgments:** These studies were supported from ADAMA-USA.

**Conflicts of Interest:** The authors declare no conflict of interest.

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
