# Peer review of "Metamitron Thinning Efficacy of Apple Fruitlets Is Affected by Different Rates, Timings and Weather Factors in New York State"

_horticulturae, doi:10.3390/horticulturae9111179_

Round 1

Reviewer 1 Report

Comments and Suggestions for Authors

Review of manuscript:

Metamitron thinning efficacy of apple fruitlets is affected by 2 different rates, timings and weather factors in New York State

The authors report a summary of a total of 18 field experiments on the effect of metamitron on thinning of Gala apples. All experiments were carried out as complete randomized block designs. The data were properly analyzed and the conclusions drawn are well justified and sound.

I only have minor recommendations that the authors may want to consider and address in a revised version:

1.       Is there any reason why all experiments were carried out in Gala?

2.       I find figure 2 hard to understand – please check whether a composite comprising your current figure as A and a new figure as B would be helpful. The new figure should plot the slope of the linear regression lines (fruit drop/rate) on the y axis as a function of diameter on the x-axis. Would that help or is this just duplicating the info in Fig. 3?

3.       Lettering of Fig. 3 is too small. Please enlarge.

4.       Correct Typo in heading 3.3.1

5.       Lettering of Fig. 4 too small.

6.       Legend of Fig. 5 – why was the red square drawn in there – please provide explanation in legend.

7.       Boxes in Fig. 7 – I am not sure that I understand the purpose of the boxes.

8.       Fig. 9. I am almost inclined to fit an asymptotic relationship here – high slope at temperatures below 21C, no effect of temperature in the range from 22 to 27°C. Would that be better?

9.       Line 342: This should probably read “than” instead of “that”…

I recommend the paper to be published with minor changes.

Author Response

Thanks for all your suggestions.
I accept and answer all your questions in the word document
Thanks

Reviewer 2 Report

Comments and Suggestions for Authors

This manuscript summarized long-term Metamitron treatment studies. The results and conclusions are generally consistent with previous studies, so I do not feel that they are particularly novel, but the results were presented in an easy-to-understand manner, and I believe that it is worth publishing. Useful information has been provided by identifying the period of solar radiation that affects the effectiveness of the Metamitron treatment.

Minor comments.

1. I would like to know when to count the number of fruits to evaluate fruit thinning efficacy. Did you count them immediately after June drop? Or was it counted at harvest time?

2. The values on the vertical axis of the figures were not expressed as percentages.

3. There were several places where authors explained the results without citing figures.

4. What is the journal in reference 38?

Author Response

Thanks for all sugections i accept and changed.

You can see my answer in red color

Thanks

  1. I would like to know when to count the number of fruits to evaluate fruit thinning efficacy. Did you count them immediately after June drop? Or was it counted at harvest time? I modify and I add this information the section 2.4. We always evaluate the thinning efficacy at harvest because if we count after natural fruit drop the fruit is small and it si easy to do mistakes.
  2. The values on the vertical axis of the figures were not expressed as percentages. Done
  3. There were several places where authors explained the results without citing figures. Done
  4. What is the journal in reference 38? Done

Reviewer 3 Report

Comments and Suggestions for Authors

The manuscript is quite well written, easy to read and understand. Apart from some minor technical correction inserted in the attached document I suggest more detailed description of the orchards where the trials were set up. Also, I suggest using term "solar radiation" instead of just "radiation".

Author Response

Thanks for all sugestions i answer in the pdf and red

I modifyed all sugections

The manuscript is quite well written, easy to read and understand. Apart from some minor technical correction inserted in the attached document I suggest more detailed description of the orchards where the trials were set up. Also, I suggest using term "solar radiation" instead of just "radiation".

I changed in all text
